# Comprehensive health risk assessment of microbial indoor air quality in microenvironments

**Pradeep Kumar[1], A. B. Singh[2], Rajeev Singh****[1]***

**1** Department of Environmental Studies, Satyawati College, University of Delhi, Delhi, India, **2** CSIR- Institute of Genomics and Integrative Biology (IGIB), Delhi University Campus, Delhi, India

\* 10rsingh@gmail.com

## Abstract

The higher airborne microbial concentration in indoor areas might be responsible for the adverse indoor air quality, which relates well with poor respiratory and general health effects in the form of Sick building syndromes. The current study aimed to isolate and characterize the seasonal (winter and spring) levels of culturable bio-aerosols from indoor air, implicating human health by using an epidemiological health survey. Microorganisms were identified by standard macro and microbiological methods, followed by biochemical testing and molecular techniques. Sampling results revealed the bacterial and fungal aerosol concentrations ranging between (300–3650 CFU/m$^3$) and (300–4150 CFU/m$^3$) respectively, in different microenvironments during the winter season (December-February). However, in spring (March-May), bacterial and fungal aerosol concentrations were monitored, ranging between (450–5150 CFU/m$^3$) and (350–5070 CFU/m$^3$) respectively. Interestingly, *Aspergillus* and *Cladosporium* were the majorly recorded fungi whereas, *Staphylococcus*, *Streptobacillus*, and *Micrococcus* found predominant bacterial genera among all the sites. Taken together, the elevated levels of bioaerosols are the foremost risk factor that can lead to various respiratory and general health issues in additional analysis, the questionnaire survey indicated the headache (28%) and allergy (20%) were significant indoor health concerns. This type of approach will serve as a foundation for assisting residents in taking preventative measures to avoid exposure to dangerous bioaerosols.

## 1. Introduction

Indoor air quality plays a vital role in the health and well-being of people as 80–90% of the total time is spent in the indoor environment [1,2]. Typically, people inhale 10m$^3$ air daily, which may be abundantly populated by various microorganisms in the form of colloidal suspension known as bio-aerosols [3]. Airborne microorganisms (AM) are cosmopolitan and even can be found in the highly controlled environment of operation theatres [4,5]. Fungi are the potential source of allergic and general health problems in different ways [6]. Sources of these microorganisms in indoor air may include people, organic dust, storage of various

**Funding:** This work was financially supported by Science and Engineering Research Board,

Department of Science and Technology (SERB-DST), New Delhi, India (Grant No. ECR/2017/000470). The funders had no role in study design, data collection and analysis, decision to publish, or preparation of the manuscript.

**Competing interests:** The authors have declared that no competing interests exist.

products, and circulation of air through natural and artificial ventilation systems [7] According to a previous study, roughly 30% of office workers suffer from ailments due to poor indoor air quality [8]. Exposure to these contaminants may cause allergic reactions, infections, intoxication, and various molecular reactions [6,9–13]. The importance of airborne microorganisms on health was highlighted by the World Health Organization (WHO) [14]. Moreover, respiratory and other health problems in office workers related to the indoor air quality of the built environment are termed as Sick Building Syndrome (SBS) [15].

Microbes can adapt themselves in different conditions hence, significant variation could be found in concentration in sub-areas of the same microenvironment [16]. Seasonal variations have been reported in the composition and abundance of AM, but these can also be influenced by temperature, relative humidity, and air exchange rate [17–19]. The presence of AM in prenursery schools may also affect the respiratory and paediatric health as well as the well-being of the children [20] In a recent study, *Aspergillus*, *Curvularia*, *Penicillium*, and *Rhizopus* were major fungal genera recorded from the different indoor areas in Kolkata, India [21]. The presence of mycotoxin-producing micromycetes in buildings plays a crucial role in causing SBS in occupants [22] Hospitals are one of the most significant indoor microenvironments for the spreading and propagation of the aero-microflora [23]. An air sampling study conducted in a Haematology hospital by Cho and co-workers reported that the *Aspergillus* and *Penicillium* were the most dominant fungal genera in the outdoors and inside buildings [24] **(Fig 1)**.

Delhi, India's national capital, is the country's most polluted city, drawing increased attention from the government and the general public. A few studies have been conducted in Delhi to assess the total bacterial and fungal concentration in residential areas and waste dumping sites [25–28] However, previous studies could not indicate the possible association between indoor air quality and health issues in different seasons.

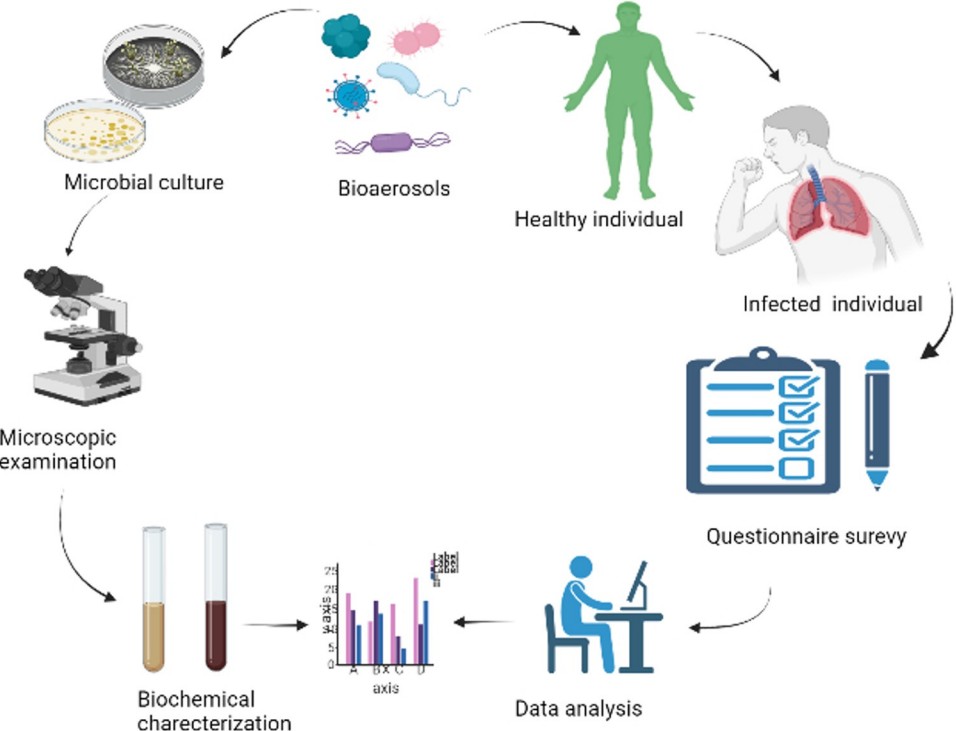

**Fig 1. Overview of the research study.**

The main objective of the present study is to investigate the pattern of morbidity among the people living in industrial, commercial, and residential due to poor air quality. Furthermore, biological sampling data obtained from indoor sites in different seasons will provide an idea to explain the seasonal adverse health problems. The current study's findings may aid in the development and implementation of preventative public health programmes, as well as the creation of recommendations aimed at creating better indoor settings.

## 2. Methodology

### 2.1. Location and characterization of sampling sites

The study was carried out in selected areas of Delhi (North Delhi), the Indian capital, with a total area of 1,484 km$^2$. Delhi is flanked on both sides by the river Yamuna, which flows from north to south and is located between 28˚12'28˚53'N and 76˚50'77˚23'E. Total culturable bacterial and fungal counts were measured in common residential microenvironments in North Delhi from December 2018- May 2019 (Fig 2).

Residential residences, college classrooms, academies, LPG godowns, and laboratories in Satyawati College's immediate vicinity were chosen for aerobiological sampling, whereas exact outdoor regions of sampling locations served as a control. The gas agency refers to LPG godowns where many cylinders are stored, and many workers continuously handle them for loading and unloading. Along with the indoor areas, biological sampling was conducted simultaneously with similar parameters for the same duration, and also background data was measured to check the source of contamination. Sampling localities were divided into three categories residential, commercial, and industrial.

### 2.2. Biological air sampling

Air samples were collected from (December 2018 to May 2019) by using the conventional settle plate method (passive gravitational method) [29]. Petri plates containing mycological

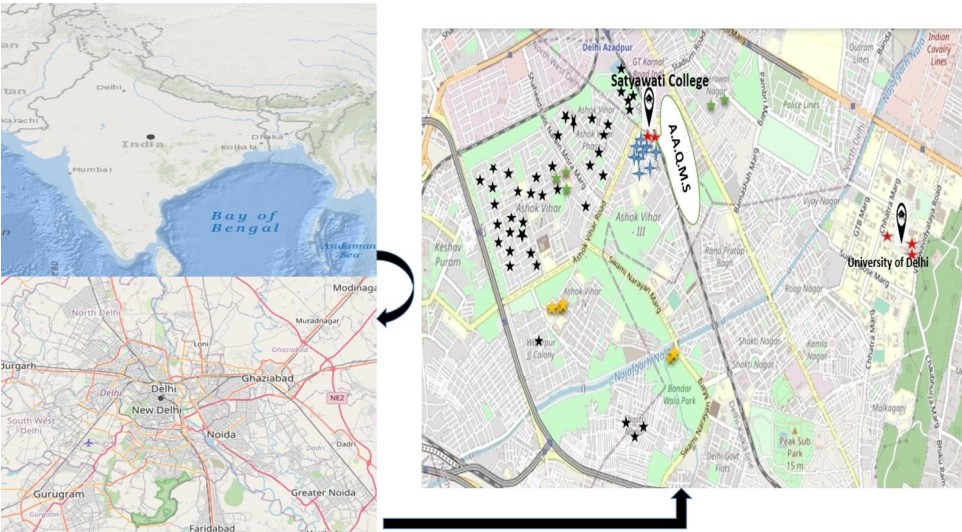

**Fig 2. Geographical location of the biological sampling areas, ambient air monitoring site (Map was generated by using USGS National Map Viewer (https://apps.nationalmap.gov/viewer) (Houses- black star, classrooms-blue star, academies-green star, laboratories-red star, and LPG godowns yellow star, (A.A.Q.M.S.)- Ambient Air Quality Monitoring System).**

medium (Sabourd Dextrose agar with Chloramphenicol and Tryptic Soy agar supplemented with cycloheximide) were used for study). All the samples were collected in triplicate and one set of agar plates was also exposed in a clean air environment as a control. Sampling was performed aseptically in each room at an average height of 1.5 m (breathing height). Petri plates were exposed for 10 minutes in case each sample.

## 2.3. Enumeration and identification of the culturable bioaerosols

Culturable bioaerosols from all the microenvironments were cultured at optimum conditions. The airborne microorganisms were counted using the plate count method. After sampling, bacterial samples were transferred to the laboratory aseptically and incubated for cultivation at 37˚C for 24–36 hours. For fungal samples, the incubation was done at 26–28˚C for 7–14 days. A digital thermometer (Checktem Digital Thermometer- HI98501) was used to assess indoor temperature and relative humidity (HTC HT-306 Temperature Humidity meter). After incubation, the bacterial and fungal colonies were analysed by standard procedures. Bacterial colonies were counted by using a colony counter. Microbial growth on agar plates was initially identified by the morphology and appearance of colonies. Furthermore, Gram's staining was performed followed by biochemical and molecular characterization in the case of bacteria [30]. Fungal samples were visualised by using Lactophenol Cotton Blue stain [31]. Microbiological analysis of the samples were carried out and all the photographs were taken using a Nikon High-Resolution Microscope under 40X, 100X, 400X, and 1000X up to genus level. Following formula was used for calculating bioaerosol concentrations in CFU/m$^3$ [29]

$$CFU/m^3 = n \text{ x } 10,000/s \cdot t/5$$

(where n is the number of colonies on the Petri plate, s is the surface of the Petri plate and t is the time of the Petri plate exposure. All the petri dishes with growth media were exposed for 10 min.)

## 2.4. Monitoring meteorological parameters

Meteorological data was obtained from the Delhi Pollution Control Committee (DPCC) Ambient Air Quality Monitoring System, Satyawati College Ashok Vihar Station, 110052, as well as from indoor sites. Temperature (˚C), relative humidity (%), and particulate matter (PM 2.5) were measured outside and within, and their link with aerospora was identified.

## 2.5. Questionnaire survey

A health survey was conducted using a set of modified questionnaires based on MM 40 questionnaire [32]. Single-stage random sampling was performed for selecting subjects from different localities in the nearby areas. Apart from residential houses, students and workers from other residential sites, i.e. laboratories, classrooms, academies, and LPG godowns were also included in the questionnaire survey. The purpose of the questionnaire was to gather information on the subjects' health and general information. The "Institutional Ethical Committee on Human Research" of Satyawati College, University of Delhi, Delhi, approved the research. The research was performed in accordance with relevant guidelines/regulations. Duly signed written consent form was obtained from all the study participants in a well-designed format. All the participants were also explained their role in the health survey and the purpose of the study. In the case of minors, consent was obtained from their parents or their guardians. A copy of the Questionnaire is attached as supplementary material (**S 2 Questionnaire form in S1 File**).

## 2.6 Data analysis

The data of bioaerosols were expressed as mean ±standard error. Statistical analysis was performed using Statistical Package for Social Sciences (SPSS 24), and Microsoft Excel 2019. All the data passed homogeneity and normality tests were evaluated by analysis of variance (ANOVA) between different groups and a probability of (p<0.05) was considered statistically significant.

# 3. Results

## 3.1. Demographics of the enrolled subjects in the study

Residents from the randomly selected sampling sites in North-west Delhi were enrolled for health survey. A total of 223 subjects signed the consent form to participate in questionnaire survey of which 153 (68.6%) were male and 70 (31.4%) females. Table 1. shows the demographic features of the participants, e.g. age, sex, and the locality.

## 3.2 Health effects of microbial and indoor air quality on subjects

**Table 2**, illustrates the distribution of respiratory and general effects with indoor air quality on the health of the subjects. Data indicates that headache was the most dominant (28%) health effect followed by allergies (20%). 13% of subjects reported frequent coughing and problem of the runny nose at the time of examination. Subjects also reported about unusual thirst (11%), burning of irritated eyes (10%), sneezing attacks (8%), and as existing problems. Female subjects are more likely to suffer from the symptoms for the reason of probably spending more time in the indoor areas.

## 3.3 Variation in seasonal microbial concentration in different sampling sites

Microbial air sampling data obtained from selected indoor sites are represented in **Fig 3**. The average culturable microbial concentration (bacteria and fungi) in winter (December to February) and spring, (March-May) is illustrated in (Fig 3A) and (Fig 3B) respectively. During the winter season, college classrooms had the highest bacterial concentration of 2453±414 CFU/ $m^3$, while LPG godowns had the highest fungal count of 2983±1106 CFU/ $m^3$, followed by coaching academies with 2961±567 CFU/ $m^3$. (**Fig 3A**). In contrast, in the spring, the average bacterial load in all microenvironments except laboratories was around 2600 CFU/ $m^3$,

Table 1. Demographic profile of the subjects involved in the study.

| Demographical features | Number of Subjects | % of Subjects |
|---|---|---|
| *Age groups* | | |
| < 18 years | 45 | 20.1 |
| 18–40 Years | 160 | 71.7 |
| > 40 years | 18 | 8.1 |
| *Gender* | | |
| Male | 153 | 68.6 |
| Female | 70 | 31.4 |
| *Locality of enrolled subjects* | | |
| Residential | 110 | 49.3 |
| Commercial | 60 | 27.0 |
| Industrial | 53 | 23.7 |

**Table 2. Prevalence of the reported health effects/symptoms due to indoor air quality.**

| Symptom category | Occurrence (%) |
|---|---|
| **Respiratory problems** | |
| Sneezing attacks | 8 |
| Nasal congestion | 8 |
| Runny nose | 13 |
| Sore or dry throat; frequent coughing | 13 |
| Tight chest; breathing difficulties (breath shortness) | 3 |
| Emphysema | 4 |
| **General Health problems** | |
| Chapped Lips | 5 |
| Unusual thirst | 11 |
| Allergy | 20 |
| Migraines | 3 |
| Headache | 28 |
| Multiple colds | 4 |
| Dry, itching, irritated, or watery eyes | 10 |

peaking at 2719±1168 CFU/ m$^3$ in residences. In the case of fungi, more variation has been noticed between different environments, and a higher concentration was observed in LPG godowns with 3713±1665 CFU/m$^3$ (**Fig 3B**). Overall, seasonal comparison of collected air samples data from different indoor sites shows indoor microbial concentration is lower in winter than spring. Unlike bacteria, fungal communities showed significantly fluctuating results in both seasons in different microenvironments. Moreover, more variation was observed in fungal and bacterial CFU in the winter season. Despite the fact that Delhi is one of the world's most polluted cities, fungal concentrations were found to be higher in all indoor places than outdoors, with the exception of research labs. However, the lowest concentrations of microorganisms were detected in laboratories in the case of bacteria and fungi in all the seasons. Bacterial concentration was highest in college classrooms, while the fungal load remained highest in

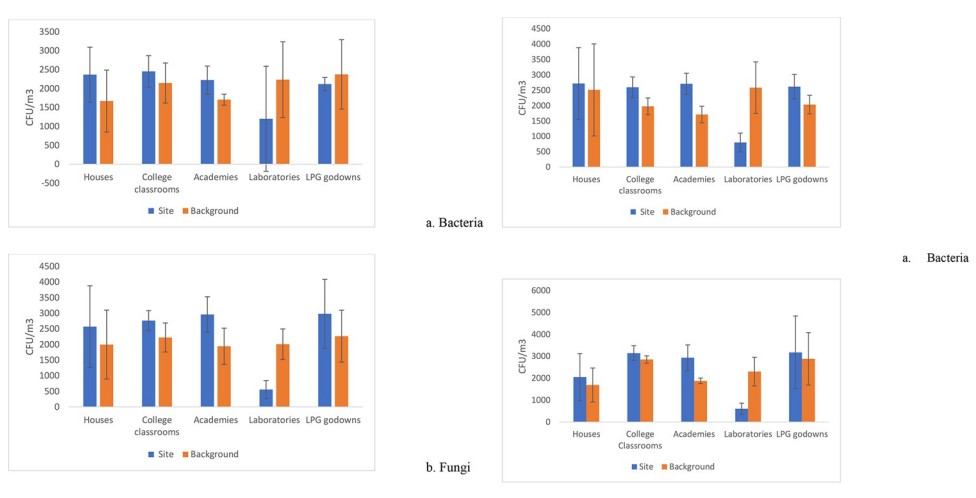

**Fig 3. a:** Average microbial concentrations of bacteria and fungi during winter season (December-February 2018–2019) Error bars show customized standard deviation. **b:** Average microbial concentrations of bacteria and fungi during spring season (Mar-May 2019) Error bars show customized standard deviation.

LPG godowns. The greater bacterial and fungal load could be due to increased activities of students and staff in academic settings compared to other locations.

## 3.4 Identification of major airborne Bacteria and Fungi present in the selected Indoor Microenvironments

All the bacterial and fungal samples were characterized by macroscopic, microscopic and biochemical methods after being culturing in laboratory. **Table 3** displays dominant culturable bacterial and fungal isolates from different indoor sampling sites. More bacterial and fungal genera were recorded from residential houses, than other sampling sites (i.e., Classrooms, LPG godowns, etc.). Of the fungal genera identified from different indoor sites shows that *Aspergillus* spp., *Penicillium*, *Rhizopus*, *Cladosporium*, *Alternaria*, and *Candida*, were present predominantly in residential houses while *Cladosporium*, *Alternaria*, and *Aspergillus* were also observed in all sampling sites. *Aspergillus* spp. and *Cladosporium* were seen common throughout all the sampling sites. *Staphylococcus*, *Micrococcus*, and *Streptobacillus* spp. were found the most dominant genera of bacteria in all the selected sites and *E. coli*, *G+ cocci*, *Pseudomonas* were also found in few sampling sites. **Fig 4** displays the photomicrographs of the dominant microbes.

## 3.5 Influence of meteorological parameters on microbial concentrations

The average temperature in all the sampling areas increased by passing of months from winter to spring (**Table 4**). December was the coldest month during sampling time, and the average temperature was highest in May. The spring season in Delhi, becomes dry and the temperature rises significantly higher in the month of April-May. Interestingly PM 2.5 concentration remained highest in December and declined to minimum in April. Relative humidity data shows fluctuation with peak in February. Indoor and outdoor microbial ratio (I/O) fluctuates

**Table 3. Major fungal and bacterial isolated in different microenvironments.**

| Name of the bacterial Isolates | Name of the Fungal Isolates |
|---|---|
| **Houses** | |
| *Staphylococcus*, | *Aspergillus* spp. |
| *Micrococcus*, | *Cladosporium* |
| *Streptococcus*, | *Penicillium* |
| *Streptobacillus*, | *Rhizopus* |
| **G–Bacilli**, | *Mucor* |
| *E. coli*, | *Alternaria* |
| *Pseudomonas*, | *Candida* |
| **G+ Cocci**, | *Candida*, |
| **Coaching Academies** | *Aspergillus* spp., |
| *Staphylococcus*, | *Fusarium* |
| **G+ Cocci**, | *Alternaria*, |
| **G+ Bacilli** | *Aspergillus* spp. |
| **LPG godowns** | *Penicillium*, |
| *Micrococcus*, | *Aspergillus* spp., |
| *Staphylococcus*, | *Fusarium*, |
| **G+ bacilli** | *Rhizopus* |
| **Research Laboratories** | *Alternaria*, |
| *Streptobacillus*, | *Aspergillus* spp. |
| *Staphylococcus* | *Candida*, |
| **College classrooms** | *Fusarium* |
| *Staphylococcus*, | |
| **G+ Cocci**, | |
| **G+ Bacilli** | |
| *Micrococcus* | |

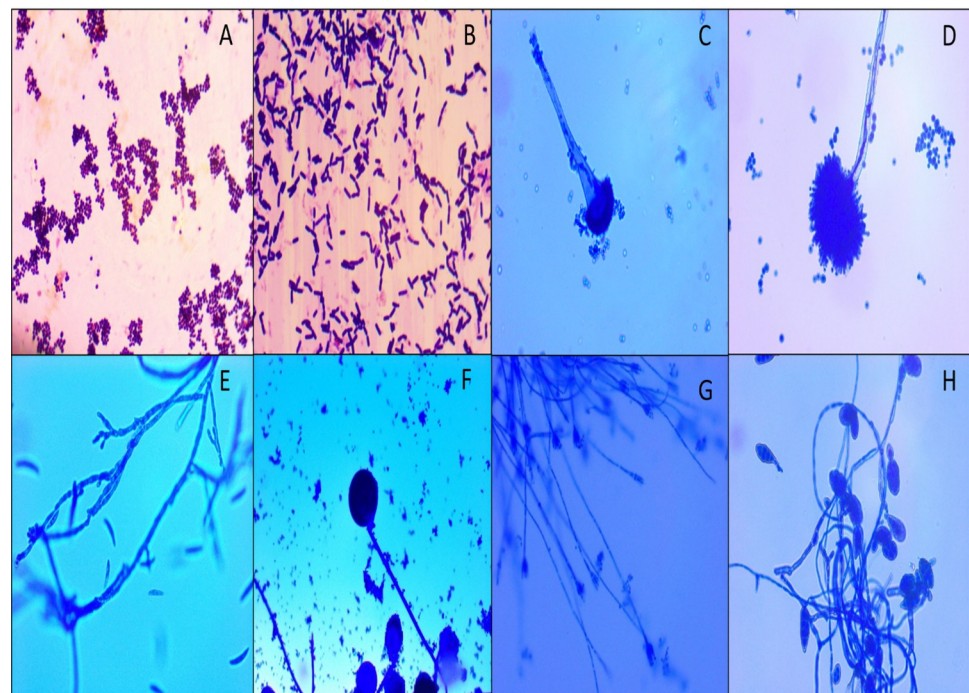

**Fig 4. Photomicrographs of dominant indoor microflora.** A. G+ *Cocci* (1000X), B. G+ *Bacilli* (1000X), C. *Aspergillus* (400X), D. *Aspergillus* (400X), E. *Fusarium* (400X), F. *Rhizopus* (400X), G. *Penicillium* (100X), H. *Alternaria* (100X).

in the case of bacterial and fungi (**Table 5**). (**Table 6**) represents correlation matrix (Paired T-test) of temperature and relative humidity with microbial concentrations in winter and spring.

## 4. Discussion

This study considered more than 65 different sampling sites (houses, classrooms, LPG godowns, laboratories, and coaching academies) for seasonal biological air sampling and health survey. Moreover, seasonal analysis of the microbes showed bacterial concentrations elevated in the houses, academies, classrooms, and LPG godowns. We observed that microbes isolated from different microenvironments belong to the opportunistic pathogens category (**Table 3**). Culturable bacterial spore count of indoor areas exceeds outdoor counts except for laboratories. In the case of fungi, indoor bioaerosols were higher than the outdoors in houses, academies, and LPG godowns. Interestingly, identical results were observed during the winter and spring seasons in the hospitals in Pakistan [33]. A previous study found that exposure to varied bacterial concentrations could have direct and indirect effects on human health [34].

**Table 4. Description of the meteorological parameters of the sampling areas (Particulate Matter PM 2.5 Data source: Ambient Air quality monitoring system (DPCC) Ashok Vihar, Delhi).**

| Season | Winter | | | Spring | | |
|---|---|---|---|---|---|---|
| Month | December | January | February | March | April | May |
| Temperature(°C) | 14.1 | 18.4 | 21.9 | 23.1 | 29.9 | 30.2 |
| Relative Humidity (RH)(%) | 63.2 | 58.7 | 63.6 | 43 | 48.4 | 36.6 |
| (PM 2.5) | 414 | 362 | 290 | 190 | 187 | 249 |

**Table 5. Indoor and outdoor microbial ratios (I/O ratio) in different sampling sites in both the seasons.**

| Site | Winter | Spring |
|---|---|---|
| **Bacteria** | | |
| Houses | 1.41 | 1.08 |
| College classrooms | 1.14 | 1.31 |
| Coaching academies | 1.30 | 1.58 |
| Laboratories | 0.53 | 0.30 |
| LPG godowns | 0.89 | 1.28 |
| **Fungi** | | |
| Houses | 1.28 | 1.21 |
| College classrooms | 1.24 | 1.10 |
| Coaching academies | 1.52 | 1.56 |
| Laboratories | 0.27 | 0.26 |
| LPG godowns | 1.31 | 1.10 |

*Staphylococcus*, *Micrococcus*, and *Streptococcus* were the most dominant bacterial genera in indoor than outdoors in the current study and previous studies [28,31,35]. A high abundance of *Cladosporium* in indoors indicate that outdoor conditions significantly affect the indoor fungal concentration. *Aspergillus* spp., *Penicillium*, *Alternaria*, *Fusarium*, and *Mucor* were the other dominant fungal genera observed indoors. These microbes may directly or indirectly play a role in respiratory diseases or other complications [11] In the current study *Aspergillus*

**Table 6. Correlation matrix of meteorological parameters temperature and relative humidity with microbial concentrations in winter and spring seasons (P<0.05 means positive correlation with meteorological parameters; significant values are in boldface.).**

| Site | Parameter | Bacterial CFU/m$^3$ | Fungi CFU/m$^3$ |
|---|---|---|---|
| **Residential Houses** | T | 0.022 | 0.030 |
| | RH | 0.023 | 0.031 |
| **College Classrooms** | T | 0.009 | 0.003 |
| | RH | 0.010 | 0.004 |
| **Academies** | T | 0.005 | 0.013 |
| | RH | 0.012 | 0.012 |
| **Laboratories** | T | 0.282 | 0.086 |
| | RH | 0.290 | 0.089 |
| **LPG godowns** | T | 0.002 | 0.045 |
| | RH | 0.002 | 0.043 |
| Winter | | | |
| Site | Parameter | Bacterial CFU/m$^3$ | Fungi CFU/m$^3$ |
| **Residential Houses** | T | 0.021 | 0.004 |
| | RH | 0.021 | 0.004 |
| **College Classrooms** | T | 0.005 | 0.003 |
| | RH | 0.005 | 0.004 |
| **Academies** | T | 0.005 | 0.013 |
| | RH | 0.005 | 0.013 |
| **Laboratories** | T | 0.046 | 0.060 |
| | RH | 0.048 | 0.061 |
| **LPG godowns** | T | 0.007 | 0.003 |
| | RH | 0.007 | 0.083 |
| Spring | | | |

spp. were recorded in all the microenvironments, the occurrence of fungal genera *Aspergillus*, *Alternaria*, *Penicillium*, and *Cladosporium* cause epidemiological allergy, allergic respiratory diseases in the inhabitants [6,9,11] however, the presence of *Alternaria* may be also responsible for asthma, skin infections, and allergic rhinitis [13]. In addition, the spore concentrations of the *Aspergillus* and *Cladosporium* were observed significantly higher in the spring season whereas, *Alternaria* was found considerably higher in the winter season [36]. A recent study by Srivastava and co-workers conducted the qualitative and quantitative analysis of bioaerosols near around Okhla landfill site Delhi [37]. Variation in the type of microbes might include industrial areas, dumping sites, slaughterhouses, waste decomposition, meteorological parameters, and anthropogenic activities [38]. Previous studies reported that the presence of *Candida* is responsible for various diseases, i.e. sinus infections, fatigue, and urinary tract infections however, presence of *Alternaria* and *Mucor* may trigger allergic reactions and asthma [9,13]. Fungi are more often released by the hair, nail, and, skin of students, which are key sources of indoor contamination of the educational buildings [39].

Previous studies confirmed that high indoor bacterial load and other physical features are responsible for the health problems in the students and staff in residential buildings [40]. A similar study conducted by Madamarandawala et al., (2019) suggested that 58% of urban and 31% of preschool students were suffering from at least one health effect related to the microbial air quality. In addition, the serum levels were elevated in children suffering from allergic reactions [41]. A similar study from Kolkata, India, also found that patients with atopic allergic history are more reactive towards fungal extracts [42]. An epidemiological study conducted in southern India, the effect of indoor air pollutants, socioeconomic, and housing characteristics on the well-being of children and women, demonstrated that these factors significantly influence respiratory health, which may increase the burden of respiratory illness [43]. Findings of the pilot study conducted in Finland to assess the indoor air problems which revealed the irritated, stuffy, or runny nose (20%), itching, burning, or irritation of the eyes (17%), and fatigue (16%) found the most common symptoms in the workers. Women had a higher proportion of indoor air problems than men, indicating the importance of the current study [12].

Few studies have been conducted in Delhi to measure biological pollution at various locations, with the majority of the studies focusing on educational institutes, libraries, dumping sites, and so on [27,28,37,44–46]. In addition, the houses of patients suffering from asthma/allergies reported (110,091) and (107,070) fungal colonies per cubic meter area in indoor and outdoor areas, respectively reported by [47]. Bacterial sampling conducted in a slum area near Jawaharlal Nehru University (South Delhi) reported a high number of bacterial aerosols between 0.43 to 3.35 x 107 CFU/m$^3$, which is significantly greater than other similar studies [27]. In another study JNU libraries recorded the total bioaerosols with range of 911–1460 CFU/m$^3$ and 2550–3110 CFU/m$^3$ for bacteria and fungi respectively [26] Ambient bioaerosol concentrations measured ranging between (1740–3224 CFU/m$^3$) for fungi and (1990–9428 CFU/m$^3$) bacteria in different areas of Delhi. Furthermore, this study also observed the most bioaerosols in stage 4 (2.1–3.3 μm), the stage 5(2.1–2.2μm), these particles are capable of penetrating the lower parts of the lungs [45].

Subjects with high microbial concentration were more likely to suffer from respiratory and general health problems. The rare respiratory disorders connected with Pseudomonas and Bacilli found mostly in residential buildings, which is related to lower and upper respiratory tract infections, was detected in the correlation between the health consequences and isolated microorganisms [48,49]. Previously, it was also observed that presence of *Bacilli* is competent in causing digestive diseases and other infections [50].

Among the prominent symptoms in respondents were burning or irritation of the eyes, as well as a regular coughing problem (**Table 2**). Previously, a study [51] suggested the presence

of microbial agents is conscientious for ophthalmic and respiratory health effects among farmers. Despite living microorganisms, spores and dead biological particles may elicit asthma and allergic reactions in the subjects [52]. According to a study conducted in south India, indoor air quality, housing conditions, and socioeconomic position are responsible for a variety of health impacts in mothers and children, including cough, runny nose, asthma, and other symptoms [43] A cross-sectional survey conducted by [53] also signified that waking up due to coughing, bronchitis, etc. were major recorded problems due to the microbial contaminated outdoor air. Similarly, the concentration of *Alternaria* was found higher in the houses of the subjects suffering from coughing. The previous report revealed that headache (20.6%), rhinitis (18.8%), and tiredness (22.1%) were common symptoms among the students moreover tiredness and ocular symptoms were found coordinated with *Aspergillus versicolor* [54]. Another cross-sectional study conducted in Australia reported the non-biological activities such as overcrowding, dust, water supply, etc., are associated with the health issues such as respiratory problems, asthma, skin problems, etc., in residents [55]. Microbial concentrations are significantly correlated with meteorological factors such as temperature and relative humidity [17,56,57]. Meteorological parameters and other growth factors are favourable in Delhi during the spring season (March-May), which could be a possible reason behind the current hypothesis. Because high temperatures are required for optimum growth, the growth of two main fungal genera, *Aspergillus* and *Penicillium*, was highest in April and May. Seasons, which are regulated by temperature, relative humidity, and exchange rates, have a substantial impact on microbial exposure [17]. The laboratories' I/O ratios ranged from 0.26–0.53, indicating that the usage of air filters and fewer human activities on this location may be responsible for the lower indoor microbial concentrations (**Table 5**). High humidity and leaking conditions promote dampness, which increases the growth of the fungi on walls and other areas [58]. Additionally, hot and dry weather increase the spore buoyancy, which causes the transport of spores from outdoor regions to longer distances [59]. Although temperature has varied impacts in each season, a prior Polish study reveals a positive association between coarse particulate matters and bioaerosol concentration [57]. Ventilation plays a crucial role in maintaining the concentrations of the airborne microflora, buildings without an air conditioner (A.C.) and other anthropogenic ventilation have 1.5 times higher concentrations of bioaerosols than with AC [60]. The concentration of airborne microorganisms in the environment is influenced by vegetation and road congestion [61]. Isolation, evaluation, and characterization are essential for controlling microbial contamination in indoor environments. Human activities, poor ventilation, and waste degradation are all major contributors of these pollutants [21]. The collected data was compared to earlier research [62,63] and it was found that the current study is highly relevant since inhabitants spend a lot of time indoors and experience a variety of difficulties connected to microbiological indoor air quality. [20,42,53,64,65]. There are good clinical and experimental results associated with dampness and elevated indoor mold concentrations [14].

Culture based technique was used in present study to assess the microbial load in indoor areas. Although the culture-based studies are reliable, effective, viable, inexpensive, but have several drawbacks in the characterization of bioaerosols. Only 1–2 percent of total ambient bacteria are culturable, implying that non-viable microorganisms make up a large fraction of the isolated microflora [66]. Culture-based methods are unable to detect unculturable microbes [67] however, culture methods have other problems, such as they are time-consuming, temperature-specific, and aseptic techniques are required to avoid cross-contamination [68]. Non-culture-based or culture-independent methods can detect a hundred to thousand times more concentration than culture-based methods [69]. Viable but non-culturable may also be underestimated in the culture-dependent techniques [70]. In the case of fungi, there is

no ideal medium for their growth also, the characteristics of fungi are influenced by several physical and chemical factors [71–73]. Some studies also recommended the morphological characteristics change in dimorphic fungi at certain temperatures [74].

## 5. Limitations of the study

The presence of high indoor bioaerosol concentrations and health effects could not be directly correlated as they may appear due to other reasons. Non-biological indoor pollutants may also cause respiratory diseases among people [75]. Volatile organic compounds emissions, $CO_2$, PM2.5 are amongst the major pollutants responsible for health consequences [64,65]. Long term exposure to environmental tobacco smoke may act synergistically with other pollutants to cause severe respiratory health effects [76]. Furthermore, cooking methods may be a cause for concern, as it has been discovered that women who are exposed to high levels of indoor air pollution are more susceptible to illness [77]. Subjects were inquired about the cooking methods and observed that people in the sampling area only use LPG stoves for cooking which has minimal effects on human health [78]. Furthermore, investigations at multiple places over longer time periods are required to provide a thorough understanding of the airborne microbiome and the various factors that influence its ecology.

## 6. Conclusion

This study suggested that high concentrations of the microbes were present in indoor areas as compared to outdoors. Indoor bacterial abundance were significantly higher in spring than winter. Headache and chronic allergies were the major symptoms perceived in inhabitants. Health survey revealed that microbial abundance is positively correlated with the meteorological parameters such as temperature and relative humidity. Human activities and ventilation facility influence the diversity, abundance, and composition of bioaerosols in indoor environments. The current study would aid in the standardization of protocols to reduce the degree of biological contamination.

## Supporting information

**S1 Dataset. Minimum data set.** The values behind the means, standard deviations and other measures reported; The values used to build graphs; The points extracted from images for analysis are provided here.
(XLSX)

**S1 File. Questionnaire form.** A copy of health proforma and subject consent form used for collecting the health data is provided here.
(PDF)

## Acknowledgments

Authors acknowledge the Delhi Pollution Control Committee (DPCC) Delhi for providing the meteorological data.

## Author Contributions

**Conceptualization:** Rajeev Singh.

**Investigation:** Rajeev Singh.

**Methodology:** Pradeep Kumar.

**Supervision:** Rajeev Singh.

**Writing – original draft:** Pradeep Kumar.

**Writing – review & editing:** A. B. Singh, Rajeev Singh.

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
