## [Decision Letter · Decision Letter 0]

19 Oct 2021

PONE-D-21-29329Comprehensive health risk assessment of microbial indoor air quality in microenvironmentsPLOS ONE

Dear Dr. Singh,

Thank you for submitting your manuscript to PLOS ONE. After careful consideration, we feel that it has merit but does not fully meet PLOS ONE’s publication criteria as it currently stands. Therefore, we invite you to submit a revised version of the manuscript that addresses the points raised during the review process.

Please pay special attention to the reviewer's comments and suggestions about providing more precise details in the methods, in addition to the reorganization of the discussion. 

We look forward to receiving your revised manuscript.

Kind regards,

Qinghua Sun, MD, PhD

Academic Editor

PLOS ONE

Journal Requirements:

2. In your Methods section, please provide additional location information about your sampling sites, including geographic coordinates for the data set if available.

3. In your Methods section, please provide additional information regarding the permits you obtained for the work. Please ensure you have included the full name of the authority that approved the sampling sites access and, if no permits were required, a brief statement explaining why.

4. Please provide additional details regarding participant consent. In the Methods section, please ensure that you have specified (1) whether consent was informed and (2) what type you obtained (for instance, written or verbal). If your study included minors, state whether you obtained consent from parents or guardians. If the need for consent was waived by the ethics committee, please include this information.

[Authors are thankful to Science and Engineering Research Board, Department of Science and Technology (SERB-DST), New Delhi, India for providing the financial support for this study. Authors are also thankful to Delhi Pollution Control Committee (DPCC) Ashok Vihar, Delhi for providing the meteorological data.]

 [Rajeev ECR/2017/000470 Science and Engineering Research Board, New Delhi https://www.serbonline.in. The funders had no role in study design, data collection and analysis, decision to publish, or preparation of the manuscript.]

7. We note that Figure 1 in your submission contain [map/satellite] images which may be copyrighted. All PLOS content is published under the Creative Commons Attribution License (CC BY 4.0), which means that the manuscript, images, and Supporting Information files will be freely available online, and any third party is permitted to access, download, copy, distribute, and use these materials in any way, even commercially, with proper attribution. For these reasons, we cannot publish previously copyrighted maps or satellite images created using proprietary data, such as Google software (Google Maps, Street View, and Earth). For more information, see our copyright guidelines: http://journals.plos.org/plosone/s/licenses-and-copyright.

a) You may seek permission from the original copyright holder of Figure 1 to publish the content specifically under the CC BY 4.0 license.  

Reviewers' comments:

Reviewer's Responses to Questions

**Comments to the Author**

1. Is the manuscript technically sound, and do the data support the conclusions?

Reviewer #1: Partly

2. Has the statistical analysis been performed appropriately and rigorously? 

Reviewer #1: No

3. Have the authors made all data underlying the findings in their manuscript fully available?

Reviewer #1: Yes

4. Is the manuscript presented in an intelligible fashion and written in standard English?

Reviewer #1: No

5. Review Comments to the Author

Reviewer #1: In my opinion, the paper needs a revision focusing on a critical review of the results and discussion in the context of the aims of the study because too many details are given as findings and thus the highlights of the research are not clearly underlined.

Despite development of active sampling of airborne bacteria and fungi, some laboratories still apply the deposition method as the basic method for the determination of the concentration of viable bioaerosol particles. Although any strictly physical relation between concentration and deposition of airborne particles does not exist, some researchers try to demonstrate that there is a high enough correlation between these two factors to enable the application of the deposition method in the bioaerosol monitoring. The reason is that this method is simple and cheap. Unfortunately, This manuscript seems to be the continuation of this philosophy. I am interested in how the concentrations in CFU/m3 were calculated using the deposition method? Could the Authors provide the formula they used when calculating bioaerosol concentrations?

Conclusions: Please clarify the conclusions. In their present form conclusions are rather a summary.

6. PLOS authors have the option to publish the peer review history of their article (what does this mean?). If published, this will include your full peer review and any attached files.

Reviewer #1: No

---

## [Author Response · Author response to Decision Letter 0]

16 Jan 2022

Manuscript number- [PONE-D-21-29329] - [EMID: af3e6b7af2fb6ff7]

Title- Comprehensive health risk assessment of microbial indoor air quality in microenvironments

Additional comments: 

Thank you for the information about Google map images. There are restrictions about Google maps which are not compatible with the Creative Commons Attribution license.

All PLOS content is published under the Creative Commons Attribution (CC BY) 4.0 license, which means that the images will be freely available online, and any third party is permitted to access, download, copy, distribute, and use these materials in any way, even commercially, with proper attribution.

We ask that you please replace your map images with ones that are acceptable to publish under the CC BY license (e.g. images in the public domain or previously published under the same CC BY license).

I hope this information has been helpful but if you have any more questions, please feel free to reach out again.

USGS National Map Viewer (http://viewer.nationalmap.gov/viewer/)

USGS Earth Resources Observatory and Science (EROS) Center (http://eros.usgs.gov/#)

The Gateway to Astronaut Photography of Earth (https://eol.jsc.nasa.gov/)

Maps at the CIA (https://www.cia.gov/library/publications/the-world-factbook/docs/refmaps.html)

NASA Earth Observatory (http://earthobservatory.nasa.gov/)

Landsat (http://landsat.visibleearth.nasa.gov/)

Natural Earth (http://www.naturalearthdata.com/)

Response: Complied

Complied: As per suggestion, satellite images has been provided by using USGS National Map Viewer (http://viewer.nationalmap.gov/viewer/) for Fig 1.

We note that your Data Availability Statement is currently as follows: "https://doi.org/10.21203/rs.3.rs-445730/v1"

We note as well that this is simply a link to your posted preprint.

Authors must share the “minimal data set” for their submission. PLOS defines the minimal data set to consist of the data required to replicate all study findings reported in the article, as well as related metadata and methods. PLOS only allows data to be made available upon request if there are ethical, legal, and/or third-party restrictions upon the dataset.

At this time, we'll require some more information from you to ensure adherence to PLOS ONE's data availability policy (https://journals.plos.org/plosone/s/data-availability). Please address all of the following points.

1. Please confirm whether there are ethical, legal, and/or third-party restrictions upon your dataset.

2. If there are no restrictions upon your minimal anonymized dataset, please either upload it as a Supporting Information file or to a stable public repository; you can find a list of PLOS-recommended repositories here (https://journals.plos.org/plosone/s/recommended-repositories).

3. If there are restrictions upon your minimal anonymized dataset, provide all of the following information.

- A complete description of the dataset.

- The nature of the restrictions (ethical, legal, third-party), and reason.

- The full name of the body imposing restrictions upon the dataset (institution, ethics committee, data access committee, etc.).

- If the data is owned by a third party, confirmation of whether the authors received any special privileges in accessing the data.

- Direct, non-author contact information (preferably email) to which data access requests can be sent. Note that it is not acceptable for an author of the study to be the sole listed point of contact for data access requests. This information will be helpful in updating your Data Availability Statement.

Response: Complied

We thank the editorial team for careful analysis of our manuscript. Currently amended data availability statement is as follows- “All relevant data are within the paper and its Supporting Information files”. Minimum data set file has been incorporated as excel file.

Journal Requirements:

Response: Complied

We thank the editor, reviewers and editorial team for careful analysis of our manuscript and pointing out the mistakes. Manuscript is being submitted as per journal format.

2. In your Methods section, please provide additional location information about your sampling sites, including geographic coordinates for the data set if available.

The additional information about sampling sites has been incorporated. Geographic coordinates for the data set of Delhi has been provided. Additionally more detailed sampling location has been mentioned in new figure including nearby areas. (page no. 5 and line no. 104-106, figure 1) 

3. In your methods section, please provide additional information regarding the permits you obtained for the work. Please ensure you have included the full name of the authority that approved the sampling sites access and, if no permits were required, a brief statement explaining why.

Response: Complied

“Institutional Ethical Committee for Human Research” of Satyawati College, University of Delhi, Delhi provided the permission for conducting the research work (S3 Ethical committee approval letter). Moreover, oral consent was obtained from the owners of the houses, Principal, Satyawati College approved the sampling sites located within the college, LPG Godowns and director of academies prior to perform air sampling. 

4. Please provide additional details regarding participant consent. In the Methods section, please ensure that you have specified (1) whether consent was informed and (2) what type you obtained (for instance, written or verbal). If your study included minors, state whether you obtained consent from parents or guardians. If the need for consent was waived by the ethics committee, please include this information.

Response: Complied

Yes 

Consent was informed and duly signed informed consent was obtained from all participants in written form. In case of minors, consent was obtained from their parents or their guardians. The research was performed in accordance with relevant guidelines/regulations. (page no. 8 and line no. 180-183)

[Authors are thankful to Science and Engineering Research Board, Department of Science and Technology (SERB-DST), New Delhi, India for providing the financial support for this study. Authors are also thankful to Delhi Pollution Control Committee (DPCC) Ashok Vihar, Delhi for providing the meteorological data.]

[Rajeev ECR/2017/000470 Science and Engineering Research Board, New Delhi https://www.serbonline.in. The funders had no role in study design, data collection and analysis, decision to publish, or preparation of the manuscript.]

Response: Complied

We thank the Editorial team for careful analysis of our manuscript and pointing out the mistake. Funding statement has amended and correct statement has been incorporated in cover letter. Please correct the funding statement on portal on our behalf. 

Response: Complied

Data of this manuscript is available as preprint. DOI of preprint is provided below. All authors confirm that this manuscript is only under consideration in Plos One Journal.

Doi: (https://doi.org/10.21203/rs.3.rs-445730/v1)

7. We note that Figure 1 in your submission contain [map/satellite] images which may be copyrighted. All PLOS content is published under the Creative Commons Attribution License (CC BY 4.0), which means that the manuscript, images, and Supporting Information files will be freely available online, and any third party is permitted to access, download, copy, distribute, and use these materials in any way, even commercially, with proper attribution. For these reasons, we cannot publish previously copyrighted maps or satellite images created using proprietary data, such as Google software (Google Maps, Street View, and Earth). For more information, see our copyright guidelines: http://journals.plos.org/plosone/s/licenses-and-copyright.

a) You may seek permission from the original copyright holder of Figure 1 to publish the content specifically under the CC BY 4.0 license. 

Response: Complied 

As per suggestion, satellite image has been omitted from the Fig 1. Alternative illustrative image showing the location of sampling areas has been supplied on the place of satellite image.

Response: Complied

As per suggestion, relevant captions for the supporting information files has been incorporated in end of the manuscript. 

Review Comments to the Author

Reviewer #1: In my opinion, the paper needs a revision focusing on a critical review of the results and discussion in the context of the aims of the study because too many details are given as findings and thus the highlights of the research are not clearly underlined.

Response: Complied 

We thank the reviewer for giving suggestion. As per suggestion results and discussion of the paper have been critically examined and rewritten as per reviewer’s comments. In current form, major highlights of the research has been underlined. A new section limitations of the study has been incorporated just after the discussion section. 

Despite development of active sampling of airborne bacteria and fungi, some laboratories still apply the deposition method as the basic method for the determination of the concentration of viable bioaerosol particles. Although any strictly physical relation between concentration and deposition of airborne particles does not exist, some researchers try to demonstrate that there is a high enough correlation between these two factors to enable the application of the deposition method in the bioaerosol monitoring. The reason is that this method is simple and cheap. Unfortunately, this manuscript seems to be the continuation of this philosophy. I am interested in how the concentrations in CFU/m3 were calculated using the deposition method? Could the Authors provide the formula they used when calculating bioaerosol concentrations?

Response: Complied 

We thank the reviewer for careful analysis of the manuscript. To ensure the exact number of microbial CFU, all the experiments were performed in triplicate form simultaneously in case of indoor sampling site and outdoor. Following formula was used for calculating bioaerosol concentrations in CFU/m3 (Moldoveanu AM 2015).

 CFU/m3 = n x 10,000/s •t/5

(where n is the number of colonies on the Petri plate, s is the surface of the Petri plate and t is the time of the Petri plate exposure. All the petri dishes with growth media were exposed for 10 min.)

(page no. 7 and line no. 153-59) 

Conclusions: Please clarify the conclusions. In their present form conclusions are rather a summary.

Response: Complied 

We thank the reviewer for excellent review of our manuscript. In revised manuscript, the conclusion section has been clarified and concised.

---

## [Editor Report · Decision Letter 1]

7 Feb 2022

Comprehensive health risk assessment of microbial indoor air quality in microenvironments

PONE-D-21-29329R1

Dear Dr. Singh,

We’re pleased to inform you that your manuscript has been judged scientifically suitable for publication and will be formally accepted for publication once it meets all outstanding technical requirements.

Kind regards,

Qinghua Sun, MD, PhD

Academic Editor

PLOS ONE
---

## [Editor Report · Acceptance letter]

16 Feb 2022

PONE-D-21-29329R1 

Comprehensive health risk assessment of microbial indoor air quality in microenvironments  

Dear Dr. Singh:

I'm pleased to inform you that your manuscript has been deemed suitable for publication in PLOS ONE. Congratulations! Your manuscript is now with our production department. 

Kind regards, 

on behalf of

Dr Qinghua Sun 

Academic Editor

PLOS ONE